# Bradycardia May Decrease Cardiorespiratory Coupling in Preterm Infants

**DOI:** 10.3390/e25121616

**Published:** 2023-12-03

**Authors:** Miguel Ángel Porta-García, Alberto Quiroz-Salazar, Eric Alonso Abarca-Castro, José Javier Reyes-Lagos

**Affiliations:** 1Center of Research and Innovation in Information Technology and Communication—INFOTEC, Mexico City 14050, Mexico; miguel.porta@infotec.mx; 2School of Medicine, Autonomous University of the State of Mexico (UAEMéx), Toluca de Lerdo 50180, Mexico; aquirozs002@alumno.uaemex.mx; 3Department of Health Sciences, Metropolitan Autonomous University-Lerma (UAM-L), Lerma de Villada 52005, Mexico; e.abarca@correo.ler.uam.mx

**Keywords:** cardiorespiratory coupling, bradycardia, apnea of prematurity, neonatal care, electrocardiogram, respiratory signal, neurodevelopment, information theory

## Abstract

Bradycardia, frequently observed in preterm infants, presents significant risks due to the immaturity of their autonomic nervous system (ANS) and respiratory systems. These infants may face cardiorespiratory events, leading to severe complications like hypoxemia and neurodevelopmental disorders. Although neonatal care has advanced, the influence of bradycardia on cardiorespiratory coupling (CRC) remains elusive. This exploratory study delves into CRC in preterm infants, emphasizing disparities between events with and without bradycardia. Using the Preterm Infant Cardio-Respiratory Signals (PICS) database, we analyzed interbeat (R-R) and inter-breath intervals (IBI) from 10 preterm infants. The time series were segmented into bradycardic (B) and non-bradycardic (NB) segments. Employing information theory measures, we quantified the irregularity of cardiac and respiratory time series. Notably, B segments had significantly lower entropy values for R-R and IBI than NB segments, while mutual information was higher in NB segments. This could imply a reduction in the complexity of respiratory and cardiac dynamics during bradycardic events, potentially indicating weaker CRC. Building on these insights, this research highlights the distinctive physiological characteristics of preterm infants and underscores the potential of emerging non-invasive diagnostic tools.

## 1. Introduction

Each year, approximately 15 million babies are born prematurely before 37 weeks of gestation. These infants are susceptible to various health issues, which are the leading cause of death in children under the age of 5 [1].

The function of the cardiac and respiratory systems is influenced and regulated by the autonomic nervous system (ANS). The ongoing activity of the ANS can result in interactions between the cardiac and respiratory rhythms, also known as cardiorespiratory interactions (CRIs). These CRIs can lead to relative synchronization between the two systems, called cardiorespiratory coupling (CRC). It is acknowledged that the respiratory rhythm takes precedence over the cardiac rhythm [2]. CRC, which denotes the interplay between the pulmonary and cardiovascular systems, is mainly correlated with efferent vagal activity and is an intrinsic process that can have advantageous impacts on the general well-being of living beings by establishing collaborative associations between the cardiovascular and respiratory systems. However, there is insufficient knowledge regarding the mechanisms that promote normal and abnormal interactions between the two systems, potentially resulting in ANS alterations in various diseases [3]. Notably, CRC decreases when there is a shift in the balance between sympathetic and vagal activity, characterized by an increase in sympathetic activity and a decrease in vagal activity, as indicated in relevant research [4]. This observation persists even in individuals who have undergone training [5].

Bradycardia, as defined by the Neonatal Resuscitation Program (NRP), refers to a neonate’s heart rate dropping below 100 beats per minute (bpm) and commonly occurs in preterm infants [6]. Among all the potential complications associated with prematurity, those related to CRC may include apnea. Apnea is typically characterized by a cessation of breathing lasting at least 20 s or a shorter pause in breathing accompanied by bradycardia (a heart rate below 100 beats per minute), pallor, or cyanosis [7]. However, premature babies often experience brief apnea episodes lasting less than 20 s due to even minor interruptions in airflow, leading to hypoxemia or bradycardia.

Preterm infants face a complex array of changes, and the issue of bradycardia within the context of CRC introduces significant concern. In this way, infant bradycardia can lead to respiratory events like apnea. These events, characterized by breathing interruptions and associated bradycardia, can result in hypoxemia and cerebral hypoperfusion, posing immediate threats to the infant’s well-being [8]. Furthermore, the long-term consequences extend to potential neurodevelopmental disorders, which are well documented in preterm infants [8,9]. Bradycardia contributes substantially to morbidity and mortality in this vulnerable population [10]. Understanding the intricate interplay between bradycardia and CRC is crucial for identifying potential risks to both short-term and long-term neurodevelopmental outcomes [10,11,12]. Authors have associated these factors with delays in neurodevelopment and numerous other factors related to prematurity associated with long-term negative consequences [13]. The incidence of bradycardia in preterm infants differs based on their gestational age, birth weight, and other risk factors, including respiratory distress syndrome, sepsis, and anemia [14,15]. Furthermore, it has been reported that there is a correlation between bradycardia, low birth weight, and gestational age [16]. Despite the progress in neonatal care, bradycardia remains a profound contributor to morbidity and mortality in premature infants [17].

Premature neonates face an increased risk of developing bradycardia due to their immature respiratory and ANS. This condition can lead to apnea episodes and periodic breathing, indicative of an underdeveloped cardiorespiratory control system [18,19]. However, some newborns born prematurely or at full term may experience a drop in heart rate during feeding, even without other signs of respiratory or gastroesophageal reflux disease. The specific role of the ANS in triggering these symptoms remains unclear. It is plausible that the decrease in heart rate during feeding is linked to increased reflex parasympathetic autonomic nervous system activity [20].

Dick et al. highlighted the interrelationship between the cardiovascular and respiratory systems, which are distinct but interconnected [21]. In recent years, integrative physiology has gained increasing attention [22], providing a broader comprehension of how physiological systems interact. Several authors have examined the importance of CRC in neonates. Joshi et al. investigated alterations in CRC among preterm infants and revealed a synchronization of heart rate acceleration and deceleration with inspiration and expiration. They emphasized the sensitivity of these changes to whether the infants were provided with kangaroo care [23]. Groot et al. studied cardiorespiratory activity and its correlation with sleep quality and duration. Their findings suggest that heart rate and respiratory frequency are linked to active and quiet sleep, while term neonates exhibit stability and stillness [24]. Similar research conducted by Lucchini et al. employed cross-spectral analysis and bivariate phase-rectified signal averaging (BPRSA) to measure the occurrence and intensity of CRC. The results of the analysis indicate that the BPRSA curve could be responsive to CRC, evaluating the lag between respiratory activity and R-R series [25].

From the literature thus far, there is an evident gap concerning the impact of bradycardia on CRC in preterm infants. Evaluating CRC is especially crucial for those infants who experience bradycardia due to an underdeveloped ANS. This study aims to assess CRC in preterm infants utilizing nonlinear methodologies grounded in information theory, including mutual information (MI) and entropy. The objective is to explore some autonomic mechanisms associated with nonlinear interactions between the cardiovascular and respiratory systems. This evaluation was performed during bradycardia events and periods devoid of bradycardia. The central hypothesis is that CRC exhibits variances between bradycardia and non-bradycardia events in preterm infants.

## 2. Materials and Methods

This exploratory study used electrocardiograms (ECGs) and respiratory signals from the Preterm Infant Cardio-Respiratory Signals (PICS) database [17]. This database contains simultaneous electrocardiograms and respiratory signals from 10 preterm infants. These signals have an approximate duration ranging from 20 to 70 h. The infants had postconceptional ages (PCA) between 29$\frac{3}{7}$ and 34$\frac{2}{7}$ weeks and weights ranging from 843 to 2100 g (Table 1). The data were collected in the Neonatal Intensive Care Unit (NICU) at the University of Massachusetts Memorial Healthcare and are available at http://physionet.org/content/picsdb/1.0.0/ (accessed on 1 August 2023).

### 2.1. Data Processing Pipeline

The complete processing of the ECG and respiratory signals is summarized in Figure 1 and was executed using MATLAB^®^ vR2022b (The Mathworks, Inc., Natick, MA, USA). Data extraction of the ten infants (Figure 1a) was carried out from both ECG and respiration records (provided in the standard WFDB format) using the WFDB Toolbox for MATLAB^®^ [26,27]. The interbeat intervals (R-R) and inter-breath intervals (IBI) time series were computed based on the annotation files accompanying the PICS database for the ECG and respiratory signals, respectively (Figure 1b).

Signal preprocessing (Figure 1c) involved several considerations. An adaptive filter was applied to the R-R and IBI time series [28] to correct for any outliers (artifacts). Since the sampling is not uniform, both time series were resampled at 4 Hz (in addition to removing high-frequency noise) to redistribute the non-equidistant samples of the R-R intervals and IBI; then, both time series were synchronous. Later, both time series were divided according to bradycardia periods. We define bradycardia as events in which the heart rate slows to less than 100 bpm (or equivalently R-R > 0.6 s) and lasts for at least two beats (>1.2 s). We categorized the data into bradycardic (B) and non-bradycardic (NB) segments based on the resampled R-R and IBI time series. We applied Inequality 1 to evaluate each reliable R-R sample:(1)$$\left(i\geq 0.6\;s\right)\;\&\&\;\left(i+\left(i+1\right)\right)\geq 1.2\;s$$
where *i* is the current R-R sample, the number of B data events is significantly less than the number of NB events, resulting in B events having much fewer time samples. Random Undersampling (RUS) was then used to balance the length of B and NB samples (for both R-R and IBI time series), extracting the necessary number of time samples from R-R and IBI during NB to equal the number of samples of R-R and IBI for the B condition. Remarkably, each time the adaptive filtering process is performed for each subject, the output has slightly different data because the algorithm of the adaptive filter [28] uses random values from [$\mu_{a}\left(n\right)-0.5\sigma_{a}\left(n\right),\mu_{a}\left(n\right)+0.5\sigma_{a}\left(n\right)$] to replace those recognized as outliers, where $\mu_{a}$ is the adaptive mean and $\sigma_{a}$ the adaptive standard deviation. This adaptive filter comprises three primary steps: (i) the identification and removal of evident recognition errors, (ii) application of the adaptive percent filter to the physiological signals, and (iii) implementation of the adaptive controlling filter. Recognition errors, such as zero-length R-R or IBI intervals and pauses, are consistently excluded from consideration based on established practices in clinical acquisition systems.

Therefore, steps from 1b to 1d were performed over 100 trials, and the input for the statistical analysis (Figure 1e) is the median. Figure 2 shows the samples extracted from the R-R and IBI time series, respectively. Since information theory measures are related to probabilities and distributions, the original data are organized into bins (similar to a histogram) and transformed into probability values, as shown in the data distribution plot in Figure 2.

### 2.2. Information Theory Measures

#### 2.2.1. Notation and Preliminaries

Information theory metrics formally define dynamics and possible interactions between systems *X* and *Y* (for bivariate cases), modeled as discrete random variables. Then, *X* and *Y* can take values from their respective alphabets $A_{x}$ and $A_{y}$ according to a probability distribution denoted by $p\left(x\right)=PrPr\;\left\{X=x\right\}\;,\;x\in A_{x}$ and $p\left(y\right)=PrPr\;\left\{Y=y\right\}\;,\;y\in A_{y}$. In [29], Shannon developed a theory based on a statistical representation of a communication system. Shannon extended the information measure introduced by Ralph Hartley and linked this concept to a probability distribution associated with the source of symbols. According to Shannon, the information *I* associated with observing a symbol with a probability of occurrence p is represented by $I\left(p\right)=log\frac{1}{p}$. Thus, the information obtained from observing a symbol depends on its probability of occurrence. In other words, the less likely the symbol is to occur, the more information is gained from observing it.

The key measure in information theory is the Shannon entropy. It quantifies the information content of a random variable *X* (it can be extended to a vector of two or more random variables) as the average ambiguity associated with its outcomes [29]. Entropy measures were calculated to characterize the irregularity of the R-R and IBI time series within both B and NB segments. Additionally, mutual information and cross-entropy were employed to assess the degree of coupling between the R-R and IBI time series across these segments. A brief description of the information above indices is presented below. For improved clarity in the subsequent sections, we will consider *X* as the R-R time series and *Y* as the IBI time series.

#### 2.2.2. Entropy

The entropy (referred to as Shannon entropy) *H*(*X*) of a discrete random variable *X* with the alphabet $A_{x}$ and a probability mass function $p\left(x\right)=PrPr\;\left\{X=x\right\}\;,\;x\in A_{x}$ is a measure of uncertainty of occurrence of a certain event, quantifying the amount of information a variable has:(2)$$H\left(X\right)=-\sum_{i=1}^{n}\;p(x_{i})\log_{2}p(x_{i})$$
where *H* is the measure of entropy, *p* is the probability of observing the *i*th value of the bin series data *x*, and *n* is the number of bins. We consider the log in base two as bits as units of entropy.

#### 2.2.3. Cross-Entropy

Also proposed by Kullback and called directed divergence [30], the cross-entropy $cH\left(X,Y\right)$ measures the degree of difference between two discrete random variables *X* and Y, with the alphabet $A_{x}$ and $A_{y},$ and a probability distribution $p\left(x\right)$ and $p\left(y\right)$, respectively:(3)$$cH\left(X,Y\right)=-\sum_{i=1}^{n}\;p(x_{i})\log_{2}p(y_{i})$$
(4)$$cH\left(Y,X\right)=-\sum_{i=1}^{n}\;p\left(y_{i}\right)\log_{2}p\left(x_{i}\right)$$

While there are numerous approaches to computing cross-entropy [29], we chose to use the widely accepted definition in Machine Learning [31] to avoid entering the debate over the optimal cross-entropy measure, which is beyond the scope of this paper.

#### 2.2.4. Mutual Information

Mutual information (*MI*) is defined as a measure of the amount of information shared between two discrete random variables *X* and *Y*, with the alphabet $A_{x}$ and $A_{y},$ and a joint probability distribution $p\left(x,y\right)$. It quantifies the nonlinear dependence of two signals. In terms of entropy, the MI can be calculated as the sum of two variable entropies minus the joint entropy [32]. Equation (5) shows the general form of MI.
(5)$$MI\left(X,Y\right)\;=\;H\left(X\right)\;+\;H\left(Y\right)\;-\;H\left(X,Y\right)$$
where $H\left(X,Y\right)$ represents joint entropy of time series *X* and *Y*, respectively.

### 2.3. Statistical Analysis

The 100 trials performed on each of the ten subjects yielded 1000 values of the entropy of R-R and IBI, cross-entropy, and mutual information (*MI*) under both B and NB events. To simplify the analysis, we computed the median of these 100 measurements for each participant in both groups (B and NB), resulting in a single value per subject in each group (*n* = 10). Given the small sample size and the uncertain distribution of the outcomes, which cannot be assumed to be approximately normal, nonparametric tests were considered appropriate [33]. Therefore, a Wilcoxon matched-pairs signed rank test for repeated measures was chosen as the statistical method to assess differences. These results were gathered and processed for within-subjects analysis using GraphPad Prism 8 (GraphPad Software, La Jolla, CA, USA) software.

## 3. Results

Figure 3 presents the boxplots representing the average values of the information-theory indices for both bradycardia (B) and non-bradycardia (NB) samples. Initially, it was essential to ensure that each set of extracted data demonstrated the presence of either bradycardia or non-bradycardia, which was established through the distribution of the R-R intervals. An example is depicted in Figure 2.

The entropy of R-R and IBI time series, *MI*, and cross-entropy, revealed significant differences between the B and NB segments. These observed differences provide evidence, indicating decreased MI and entropy values in the B segments compared to the NB segments.

Focusing on the entropy of R-R, as illustrated in Figure 3a, the B segments were noted to have significantly lower mean values than the NB segments (3.656 ± 1.052 vs. 4.343 ± 0.9411, respectively, *p* = 0.0371). In a similar vein, the analysis of the entropy of IBI displayed in Figure 3b showed a significant decrease in mean values for the B segments in comparison to the NB segments (3.950 ± 0.7259 vs. 4.358 ± 0.8800, respectively, *p* = 0.0244).

Upon examining the mutual information indices (Figure 3c), a significant difference was identified between the B and NB segments. The B segments exhibited an average MI value of 0.3895 ± 0.1184, while the NB segments exhibited a higher average value of 0.6582 ± 0.3110. This significant difference, with a *p*-value of 0.0137, is visually represented in Figure 2c.

In contrast, the cross-entropy (R-R -> IBI and IBI -> R-R) did not reveal significant differences between the B and NB segments. The calculated average cross-entropy (R-R -> IBI) for B and NB was 5.804 ± 2.944 and 5.640 ± 2.531, respectively, resulting in a *p*-value of 0.3477. Similarly, the cross-entropy (IBI -> R-R) for B and NB was 6.701 ± 3.923 and 6.272 ± 2.766, respectively, resulting in a *p*-value of 0.500.

## 4. Discussion

While most of the research concerning bradycardia in preterm infants has been centered on predicting bradycardic/apneic events before their occurrence [17,34,35,36], our study stands out as the exploratory effort in assessing CRC in preterm infants during events, whether they involve bradycardia or not. Consequently, the significance of this preliminary study lies in its contribution to advancing our comprehension of the intricate physiological cardiorespiratory mechanisms through information theory tools. Moreover, it helps us recognize potential alterations in their complex dynamics and nonlinear correlations, offering valuable insights into this crucial aspect of infant healthcare. This exploratory study was designed to elucidate further whether a notable difference exists in the interaction between the cardiovascular and respiratory systems among preterm infants during episodes with and without bradycardia. The rationale behind this investigation stems from the understanding that premature birth can result in underdeveloped coordination between these systems, with bradycardia being a potential indicator of immature control over cardiovascular and respiratory functions [37,38]. 

Our study’s initial results indicate that the interplay between cardiac and respiratory systems may be altered during bradycardia episodes compared to non-bradycardic periods. Complexity measures applied to the R-R and IBI time series demonstrate a significative decrease during bradycardic episodes, as illustrated in Figure 3a,b, corresponding to reduced entropy. This observation is in line with prior studies that have examined abnormal neonatal cardiac conditions [39,40]. The noted reduction in complexity supports the hypothesis that pathological states, such as congestive heart failure, are characterized by less complexity in their physiological patterns than healthy systems [41]. Considering these findings, the link between bradycardia and a decreased complexity in the cardiac and respiratory systems—as suggested by the lower entropy values in R-R and IBI segments during bradycardic periods—warrants careful interpretation. Our study does not definitively establish causality. Furthermore, the lack of significant difference in cross-entropy regarding the influence of respiration on R-R intervals (or the inverse) highlights the need for a more detailed investigation into their interconnection. Subsequent research is essential to elucidate the underlying mechanisms and to potentially confirm a causal relationship between the complexity of the cardiac and respiratory systems and the incidence of bradycardia in preterm infants.

Mutual information, derived from information theory, measures the potential nonlinear interactions between two variables—here, the R-R intervals of the heart and the IBI from respiration. The *MI* values observed in the NB segments could suggest nonlinear associations between the cardiac and respiratory systems when bradycardia is absent. Such a relationship might imply a level of synchronization between the heart and lungs that could be important for physiological stability. Conversely, the *MI* values in the B (bradycardic) segments may reflect a different degree of interaction during bradycardic episodes, which the nature of bradycardia itself might influence. The slower heart rate associated with bradycardia could potentially affect the usual synchrony between the cardiovascular and respiratory systems, possibly leading to a change in their capacity to adjust and communicate effectively with one another, a process that is crucial for maintaining balance within the body and adapting to changes in the environment [3]. However, these interpretations remain speculative, and the results do not definitively establish a difference in cardiorespiratory coupling (CRC) between bradycardic and non-bradycardic states. Further investigation is needed to understand these associations fully.

In light of these speculative interpretations, it is well recognized that the heart and lungs are engaged in a constant reciprocal interaction, forming a functional and anatomical reserve that underpins CRC. This dynamic exchange is key to the seamless coordination between these systems under typical conditions. Disruptions in this coupling may manifest in the face of various cardiac or pulmonary disorders, including bradycardia. Therefore, the findings of our study, which hint at altered interactions during bradycardic episodes, could indicate such disruptions. This notion aligns with the established concept of ongoing interaction between the heart and lungs, yet the exact relationship with bradycardia observed here remains hypothetical, underscoring the need for further research to clarify the nature of this linkage.

These findings underscore the importance of comprehending in detail the dynamics of the interaction between the heart and lungs in premature infants, as this understanding can have significant implications for the health and neurodevelopment of this population. Identifying and addressing alterations in this interaction could help prevent or mitigate issues and delays in the neurodevelopment of premature infants [12]. Furthermore, this research emphasizes the need to consider complex and nonlinear factors when assessing the health of premature infants, as these factors can substantially impact their short-term and long-term well-being.

Building on this, it is widely acknowledged in existing research that preterm infants are at a higher risk for various health issues and suboptimal neurodevelopmental outcomes, the mechanisms of which remain incompletely understood. A related study evaluated the autonomic development of newborns based on their gestational age at birth, with a specific focus on regulating cardiorespiratory functions. The research suggests that infants born prematurely at 35–36 weeks of gestational age exhibit a higher susceptibility to breathing instability, evidenced by diminished CRC [42]. It underscores the observation that an increase in gestational age at birth corresponds to an enhanced coupling between the two systems. These findings highlight the lower level of maturation in preterm infants and their underdeveloped cardiorespiratory regulation, which aligns with epidemiological data and may indicate a heightened risk for divergent outcomes [42]. Given the exploratory nature of our research, it suggests a direction for future studies to investigate further and possibly expand upon the understanding of cardiorespiratory interactions in preterm infants, with a particular focus on bradycardia. A study utilizing the same database for predicting bradycardic events revealed that increased variance in the heart rate signal was a precursor to severe bradycardia. This increased variance was associated with a rise in power from low content dynamics in the low-frequency and diminished multiscale entropy values preceding bradycardia [17]. This observation amplifies our findings, pointing towards nuanced physiological shifts before the onset of bradycardic episodes. Notably, previous findings have shown a significant negative correlation between entropy and the duration of respiratory support required, further emphasizing the potential clinical implications of these observations [43].

Furthermore, additional research has explored the ramifications of initial immunization on cardiorespiratory events in extremely preterm infants. The observed cardiorespiratory events were mainly attributed to a dominant sympathetic influence over heart rate, reduced Heart Rate Variability (HRV), characterized by low entropy, and a continuing immaturity in controlling respiratory rhythm [44]. These findings reiterate the complexity of the regulatory mechanisms in preterm infants and their susceptibility to alterations in physiological states. Another significant contribution to this discourse was a study highlighting the predictive capabilities of cardiorespiratory variability parameters during non-invasive respiratory support in extremely preterm infants. The study demonstrated moderate predictive accuracy for successful extubation, with these measures showcasing notable differences between successful and unsuccessful extubation cases. This highlights their potential as biomarkers to assess extubation outcomes in this vulnerable group [45].

The uniqueness of our work lies in its application of information theory tools to assess alterations in CRC from recorded data, which may reveal patterns that are not immediately apparent in predictive models. This retrospective analysis could inform the development of sophisticated monitoring techniques that, while not real-time, could still enhance clinicians’ ability to understand and respond to bradycardic episodes or other cardiorespiratory anomalies. Additionally, by retrospectively assessing CRC during bradycardia, our study offers fresh insights into the physiological states of preterm infants. We deepen our understanding of the cardiorespiratory system’s function during these critical periods by examining the reduced complexity and potential nonlinear correlations between cardiac and respiratory signals. This knowledge could lead to improved clinical strategies to enhance this vulnerable population’s health and developmental outcomes.

The current study represents a novel approach in understanding the interplay between bradycardia and CRC in preterm infants. Previous studies in this field, such as those by Faes et al., 2014 [46], Rozo et al., 2021 [47], Lucchini et al., 2020 [48], and Lucchini et al., 2018 [25], have laid important groundwork. Faes et al.’s study focused on the use of model-free tools for time series analysis to understand physiological system interactions, introducing a method to evaluate the direction, magnitude, and timing of information transfer between systems. Rozo et al. compared various methods to estimate Transfer Entropy (TE) in cardio-respiratory interactions, finding adaptive partitioning most effective. Their work emphasized the importance of choosing appropriate signals and methods for analyzing such interactions. The study by Lucchini et al., 2020, explored cardiorespiratory information transfer in healthy neonates, aiming to describe its development relative to gestational age. They extended the traditional TE measure to analyze both instantaneous and delayed effects between cardiac and respiratory systems. Lastly, Lucchini et al., 2018, investigated the phase coupling and its directionality in newborn infants, assessing the influence of gestational age at birth on the development of this synchronization.

The current study builds upon these foundations by specifically focusing on the differences in CRC during bradycardic and non-bradycardic events in very preterm infants. Unlike the previous studies which primarily involved healthy subjects or general neonate populations, this research delves into a more vulnerable group, very preterm infants experiencing bradycardia. The use of the PICS database for analyzing cardiac and respiratory time series, coupled with information theory measures, marks a distinctive approach. This study’s findings about the lower entropy values in bradycardic segments and the implications for reduced complexity in cardiorespiratory dynamics during such events contribute significantly to the understanding of autonomic maturation and the interplay between cardiac and respiratory systems in this high-risk population. This direction is crucial for developing better diagnostic tools and enhancing healthcare outcomes for preterm infants.

CRC assessment may help provide additional measures of infant well-being, which would be complementary in a clinical setting. To follow up these investigations in the future, CRC features characterized by information theory measures could be used to implement classifiers to distinguish bradycardic periods from regular ones. Although at first glance, it may seem redundant to develop classifiers to distinguish bradycardia, as it can be easily identified through heart rate, there are merits in considering this approach. Bradycardia can be readily detected, but identifying subtle patterns, correlations, and nonlinear features that could indicate the severity, etiology, or prognostic implications of bradycardia may not be as straightforward. The classifiers developed from information theory measures could identify these features and provide a deeper understanding and earlier detection of cardiorespiratory irregularities. This could justify the effort and resources needed to develop and implement such tools in a clinical setting.

Despite the limitations inherent in using electrocardiograms and respiratory signals, these remain standard and widely accepted methods for assessing CRC [49,50,51]. Advances in technology have improved the accuracy of these measurements, and the use of signal-processing techniques can mitigate the impact of artifacts. Furthermore, complementary methods and additional physiological measures can be incorporated to validate findings and ensure the robustness of the results.

Finally, these novel measures might become non-invasive complementary diagnostic tools to investigate the physiological mechanisms involved in bradycardia and potential predictors of bradycardic periods. Future work consists of the computation of relevant indexes such as functional autonomic age (FAA) derived from the ECG signal [42] and respiratory signs; it offers direct avenues towards estimating autonomic maturation at the bedside during intensive care monitoring.

## 5. Conclusions

This preliminary study provides an exploratory look at cardiorespiratory coupling in preterm infants, comparing episodes with and without bradycardia. The changes noted in the entropy of interbeat (R-R) and inter-breath (IBI) interval time series might reflect a different organization in the respiratory and cardiac dynamics during bradycardic events. Moreover, the observed variations in mutual information may suggest potential changes in the cardiorespiratory interaction during such events. Our results could imply a reduction in the complexity of respiratory and cardiac dynamics during bradycardic events, potentially indicating weaker CRC. These initial observations contribute to a nuanced understanding of the physiological intricacies of preterm infants and highlight the possible utility of new metrics as non-invasive diagnostic tools. The noted variations in entropy raise questions about possible neurodevelopmental concerns in preterm infants. These preliminary findings emphasize the importance of further research to explore the broader implications of cardiorespiratory interactions on these infants’ neurodevelopment and overall health. The study suggests directions for future research that may advance our understanding of autonomic growth and cardiorespiratory relationships, ultimately aiming to improve healthcare outcomes for preterm infants.

## Figures and Tables

**Figure 1 entropy-25-01616-f001:**
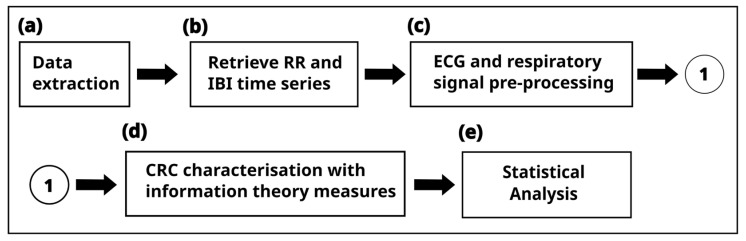
Cardiorespiratory signal processing pipeline.

**Figure 2 entropy-25-01616-f002:**
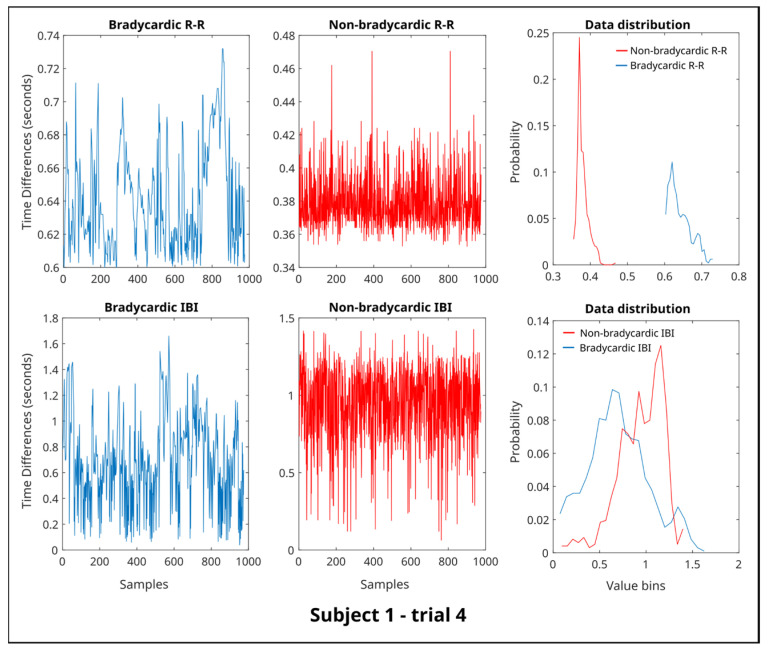
Example of the data distribution for both interbeat intervals (R-R) and inter-breath intervals (IBI) time series during bradycardic (B) and non-bradycardic (NB) conditions (Subject 1, trial 4).

**Figure 3 entropy-25-01616-f003:**
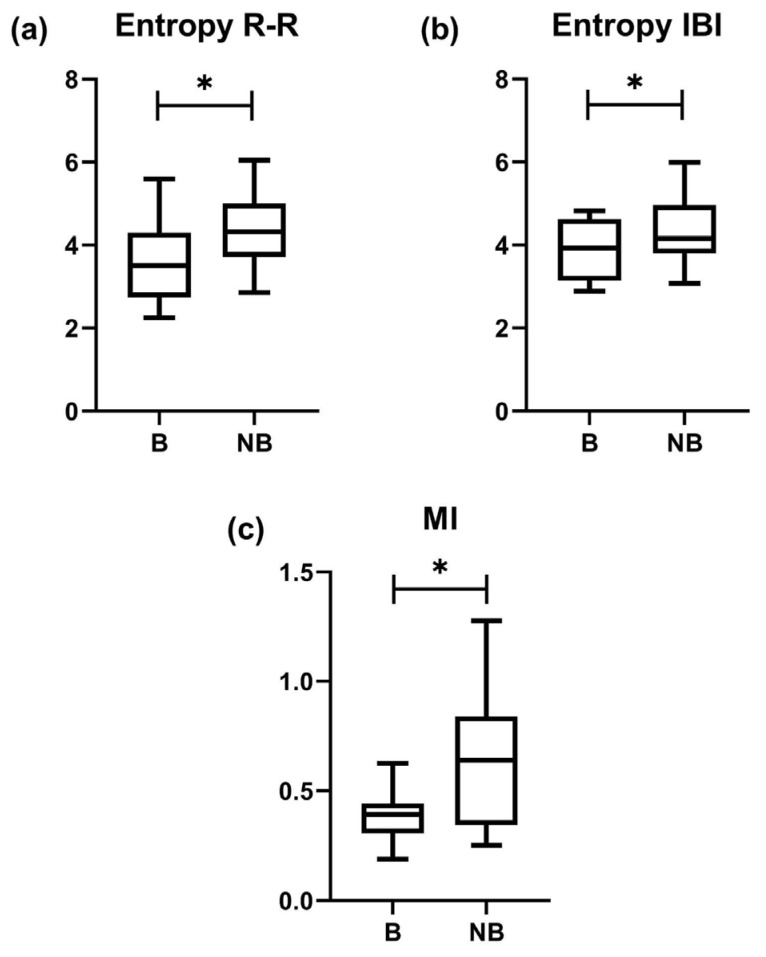
Boxplots of significant information-theory indices related to complexity between interbeat intervals (R-R) and inter-breath intervals (IBI) segments of preterm infants during bradycardia (B) and non-bradycardia events (NB). (**a**) Entropy of R-R time series between B and NB; (**b**) Entropy of IBI time series between B and NB; and (**c**) Mutual information (MI) of R-R and IBI between B and NB. * *p* < 0.05 between B and NB by Wilcoxon matched-pairs signed rank test.

**Table 1 entropy-25-01616-t001:** PCA, weight, and mean heart rate of PICS subjects.

Subject	1	2	3	4	5	6	7	8	9	10
PCA	$29\;\frac{3}{7}$	$30\;\frac{5}{7}$	$30\;\frac{5}{7}$	$30\;\frac{1}{7}$	$32\;\frac{2}{7}$	$30\;\frac{1}{7}$	$30\;\frac{1}{7}$	$32\;\frac{3}{7}$	$30\;\frac{4}{7}$	$34\;\frac{2}{7}$
Weight (kg)	1.2	1.76	1.71	0.84	1.67	1.14	1.11	2.1	1.23	1.9
Mean heart rate (bpm)	155	131	131	167	143	137	162	141	150	156

This table is an extract from [10].

## Data Availability

Data are contained within the article.

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
