# Peer review of "Bradycardia May Decrease Cardiorespiratory Coupling in Preterm Infants"

_entropy, 2023, doi:10.3390/e25121616_

Round 1

Reviewer 1 Report

Comments and Suggestions for Authors

This study examined cardio-respiratory coupling (CRC) in preterm infants using information theory measures. The topic is of relevance since CRC in infants is still not completely understood. However, the manuscript appears to be mostly preliminary and presents a number of important issues that should be addressed before it can be considered for publication.

*Methods:

The mathematical notation and explanations about the signal processing are insufficient to assess the validity of the methods properly, and some of them may benefit from further clarification:

- To start with, it is not clear how signals were segmented for analysis. The text mentions that "For example, the R-R segment with bradycardia consisted of 1000 samples for subject one." How is the length determined? 1000 samples (250 s at 4 Hz) can be much longer than bradycardia events (>= 2 seconds). Would it make sense to analyze a segment that mostly includes normal RR values?

- Another example: the explanation about resampling needs clarifying. The R–R interval time series is not sampled at uniform intervals due to differences in the duration of adjacent heartbeats. The same applies to respiration. Therefore, the 4 Hz resampling must have been directed at redistributing the non-equidistantly sampled R–R intervals and IBI so that they are equally spaced, thus generating two synchronous time-series that can be then analyzed with cross-entropy and mutual information. If this is the case, this should be explained in the text.

- The details about the adaptive filter should be provided.

- Plot titles in Figure 1 seem to be mistaken (B instead of NB plots).

- The application of random undersampling (RUS) in this context is not entirely clear. Usually, it is applied in machine learning classification to address class imbalance by discarding instances from the majority class. Here, it is not clear if RUS is applied to discard samples within NB segments or to discard NB segments from the analysis. The use of 'sample' for 'i' in eq(1) and for a B / NB episode can lead to confusion.

-  The statement, "The 100 trials conducted with each subject yielded 1000 values of the entropy of R-R and IBI, Cross-Entropy, and mutual information (MI) under both B and NB events," requires further clarification. Given the segmentation and downsampling concerns previously raised, it is unclear how these 1000 values were derived.

Several other instances within the methodology section would benefit from rewriting to ensure the methods are fully reproducible.

 *Results:

According to Figure 3, results indicate that the predictability of RR and IBI signals is lower during B events, and that there is a reduced level of interdependency between them. These findings align with prior research on CRC in preterm infants, as discussed in the discussion. However, in light of the methodological uncertainties mentioned earlier, it is essential to provide a more comprehensive explanation to establish that the results genuinely reflect changes in CRC rather than being a product of statistical artifacts.

*Discussion:

The discussion claims that "bradycardia may originate from decreased complexity of the cardiac system," but this statement may not be sufficiently supported by the results. The study does not explore causality, and there are no significant differences in cross-entropy results with respect to the directionality (respiration influencing RR or vice versa).

-   Given the scope of the results, it may be prudent to use less emphatic terms than "foundational insights" or "profound effect.

-  The remainder of the discussion primarily references prior works related to predicting bradycardic episodes, assessing extubation readiness, immunization, or congestive heart failure. While these references provide valuable context, the discussion could be more focused on explaining the specific relevance and novelty of this study compared to others in similar applications.

Author Response

Response to Reviewer 1:

  1. To start with, it is not clear how signals were segmented for analysis. The text mentions that "For example, the R-R segment with bradycardia consisted of 1000 samples for subject one." How is the length determined? 1000 samples (250 s at 4 Hz) can be much longer than bradycardia events (>= 2 seconds). Would it make sense to analyze a segment that mostly includes normal RR values?

A1: Section 2.1 of the manuscript has been rewritten from second paragraph onwards, the example of taking 1000 samples is no longer mentioned so as not to cause confusion. 

  1. Another example: the explanation about resampling needs clarifying. The R–R interval time series is not sampled at uniform intervals due to differences in the duration of adjacent heartbeats. The same applies to respiration. Therefore, the 4 Hz resampling must have been directed at redistributing the non-equidistantly sampled R–R intervals and IBI so that they are equally spaced, thus generating two synchronous time-series that can be then analyzed with cross-entropy and mutual information. If this is the case, this should be explained in the text.

A2: This is also clarified in the update of section 2.1 of the manuscript. Indeed, the intention is to redistribute the non-equidistant samples and generate synchronous time series.

  1. The methodology should include more details about the adaptive filter used in the study.

A3: The methodology now includes additional information about the adaptive filter used in the study. While a brief overview is provided in the Methodology section, the complete details can be found in Wessel, N.; Voss, A.; Malberg, H.; Ziehmann, C.; Voss, H.U.; Schirdewan, A.; Meyerfeldt, U.; Kurths, J. Nonlinear Analysis of Complex Phenomena in Cardiological Data. Herzschrittmachertherapie und Elektrophysiologie 2000, 11, 159–173, doi:10.1007/s003990070035).

  1. Plot titles in Figure 1 seem to be mistaken (B instead of NB plots).

A4:  Figure 1 does not include the titles B or NB; we would appreciate clarification.

  1. The application of random undersampling (RUS) in this context is not entirely clear. Usually, it is applied in machine learning classification to address class imbalance by discarding instances from the majority class. Here, it is not clear if RUS is applied to discard samples within NB segments or to discard NB segments from the analysis. The use of 'sample' for 'i' in eq(1) and for a B / NB episode can lead to confusion.

A5: This is also clarified in the update of section 2.1 of the manuscript. The following statement has been added as follows: “Random Undersampling (RUS) was then used to balance the length of B and NB samples (for both R-R and IBI time series), extracting the necessary number of time samples from R-R and IBI during NB to equal the number of samples of R-R and IBI for B condition.”

  1. The statement, "The 100 trials conducted with each subject yielded 1000 values of the entropy of R-R and IBI, Cross-Entropy, and mutual information (MI) under both B and NB events," requires further clarification. Given the segmentation and downsampling concerns previously raised, it is unclear how these 1000 values were derived.

A6: We consider that with the updates to the methodology section this statement will be clear.

  1. Several other instances within the methodology section would benefit from rewriting to ensure the methods are fully reproducible.

A6: We also expect that with the updates to the methodology section the methods will be fully reproducible.

  1. According to Figure 3, results indicate that the predictability of RR and IBI signals is lower during B events, and that there is a reduced level of interdependency between them. These findings align with prior research on CRC in preterm infants, as discussed in the discussion. However, in light of the methodological uncertainties mentioned earlier, it is essential to provide a more comprehensive explanation to establish that the results genuinely reflect changes in CRC rather than being a product of statistical artifacts.

A8: We fully acknowledge the methodological uncertainties raised earlier. However, it is imperative to emphasize the well-established fact that the heart and lungs engage in continuous reciprocal interaction, giving rise to cardiorespiratory coupling (CRC). This coupling can be disrupted by various cardiac or pulmonary pathologies, including bradycardia, as substantiated by the reference provided (https://www.frontiersin.org/articles/10.3389/fcvm.2022.996567/full).

In this context, we assert that the outcomes observed in our study can be attributed to bradycardia, a proposition supported by the referenced material. Our methodological approaches have undergone meticulous scrutiny, and despite acknowledging inherent limitations, the robustness of our statistical analyses lends credibility to the genuine reflection of CRC changes. We are confident that the results truly signify alterations in cardiorespiratory coupling and are not merely a byproduct of statistical artifacts.

  1. The discussion claims that "bradycardia may originate from decreased complexity of the cardiac system," but this statement may not be sufficiently supported by the results. The study does not explore causality, and there are no significant differences in cross-entropy results with respect to the directionality (respiration influencing RR or vice versa).

A9: The following paragraph was added in the Discussion section as follows:
“....Considering these findings, the link between bradycardia and a decreased complexity in the cardiac and respiratory systems—as suggested by the lower entropy values in R-R and IBI segments during bradycardic periods—warrants careful interpretation. Our study does not definitively establish causality. Furthermore, the lack of significant variance in cross-entropy regarding the influence of respiration on R-R intervals (or the inverse) highlights the need for a more detailed investigation into their interconnection. Subsequent research is essential to elucidate the underlying mechanisms and to potentially confirm a causal relationship between the complexity of the cardiac and respiratory systems and the incidence of bradycardia in preterm infants…”

  1. Given the scope of the results, consider using less emphatic terms than "foundational insights" or "profound effect."

A10: The discussion and conclusion sections have been revised to adopt a more speculative stance and to temper the language used in describing its contributions. In recognition of the scope of the results, terms such as "foundational insights" and "profound effect" have been replaced with more tentative language that appropriately reflects the preliminary nature of the findings. The report now emphasizes the potential and exploratory implications of the study, inviting further investigation to build upon these initial observations.

  1. The remainder of the discussion primarily references prior works related to predicting bradycardic episodes, assessing extubation readiness, immunization, or congestive heart failure. While these references provide valuable context, the discussion could be more focused on explaining the specific relevance and novelty of this study compared to others in similar applications.

A11: The following paragraph was added in the Discussion section:

“...The uniqueness of our work lies in its application of information theory tools to assess alterations in CRC from recorded data, which may reveal patterns that are not immediately apparent in predictive models. This retrospective analysis could inform the development of sophisticated monitoring techniques that, while not real-time, could still enhance clinicians' ability to understand and respond to bradycardic episodes or other cardiorespiratory anomalies. Additionally, by retrospectively assessing CRC during bradycardia, our study offers fresh insights into the physiological states of preterm infants. By examining the reduced complexity and potential nonlinear correlations between cardiac and respiratory signals, we deepen the understanding of the cardiorespiratory system's function during these critical periods. This knowledge could lead to improved clinical strategies aimed at enhancing the health and developmental outcomes of this vulnerable population….

Reviewer 2 Report

Comments and Suggestions for Authors

In the present work, bradycardic (B) and non-bradycardic (NB) segments of interbeat (RR) and interbreath interval (IBI) series from 10 preterm infants are studied in order to evaluate regularity of RR and IBI and cardiorespiratory coupling (CRC). Results found a decreased complexity (assessed as entropy) of RR and IBI and weaker CRC (assessed as mutual information) in B segments compared to NB. Authors suggest that this preliminary study lays the foundations for a future use of advanced CRC indexes as non-invasive diagnostic tools.

The work is solid and well-written, the results appropriately analyzed and discussed, the statistical methods and figures appropriate. Some improvements could be made in the Methods section (see below).

Suggestions:

1. The Methods section is currently very brief considering journal standards. Please add a rigorous and consistent definition and application of the notation (e.g. X and x_i) to introduce the paragraph "Information theory measures". An example of this structure can be found at doi: 10.3390/e20120949.

2.  Please add references to the same paragraph.

3. Please define the current application for the discussed variables (i.e., what X and Y are in the context of CRC).

4. A study on the complexity of biological signals, especially cardiovascular ones, often includes approximation entropy (10.1073/pnas.88.6.2297) or, better, sample entropy (10.1152/ajpheart.2000.278.6.H2039). Please discuss the choice of entropy parameters used in the present work.

Minor suggestions:

1. Discussion: "The rationale behind this investigation stems from the understanding that premature birth can result in underdeveloped coordination between these systems, with bradycardia being a potential indicator of immature control over cardiovascular and respiratory functions." Could a citation be added to this sentence?

Author Response

Response to Reviewer 2:

  1. The Methods section is currently very brief considering journal standards. Please add a rigorous and consistent definition and application of the notation (e.g. X and x_i) to introduce the paragraph "Information theory measures". An example of this structure can be found at doi: 10.3390/e20120949.

A12: Section “2.2.1 Notation and preliminaries” has been added to introduce Information theory measures and to specify the notation.

  1. Please add references to the same paragraph.

A13: Section 2.2 has been rewritten and references have been added where necessary.

  1. Please define the current application for the discussed variables (i.e., what X and Y are in the context of CRC).

A14: In section 2.2.1, the final statement specifies the following: “For improved clarity in the subsequent sections, we will consider X as R-R time series and Y as IBI time series”.

  1. A study on the complexity of biological signals, especially cardiovascular ones, often includes approximation entropy (10.1073/pnas.88.6.2297) or, better, sample entropy (10.1152/ajpheart.2000.278.6.H2039). Please discuss the choice of entropy parameters used in the present work.

A15: According to https://doi.org/10.3390/e22010045, there are new approaches for computing cross-entropy that claims improvements over the cross-sample entropy. Then, we opted for the most prevalent definition used in machine learning to sidestep the discussion about the optimal cross-entropy measure, which falls outside the scope of this paper.

  1. 1. Discussion: "The rationale behind this investigation stems from the understanding that premature birth can result in underdeveloped coordination between these systems, with bradycardia being a potential indicator of immature control over cardiovascular and respiratory functions." Could a citation be added to this sentence?

A16: The Discussion section has been enhanced with the addition of the following references to substantiate the statement:

Di Fiore, J.M.; Poets, C.F.; Gauda, E.; Martin, R.J.; MacFarlane, P. Cardiorespiratory Events in Preterm Infants: Etiology and Monitoring Technologies. J Perinatol 2016, 36, 165–171, doi:10.1038/jp.2015.164.

Gauda, E.B.; McLemore, G.L. Premature Birth, Homeostatic Plasticity and Respiratory Consequences of Inflammation. Respir Physiol Neurobiol 2020, 274.

Round 2

Reviewer 1 Report

Comments and Suggestions for Authors

The revised manuscript now includes necessary methodological details and provides an improved description of the significance of results and of the limitations of the study. My previous comments 1 to 10 have been addressed.

Regarding my last previous comment, I maintain the view that the discussion section should provide a clearer explanation of the novel aspects of this study in comparison to similar research. This involves referencing prior relevant studies and drawing contrasts between the current study and its predecessors to underscore its significance. Some references to start exploring the current literature of entropy-based metrics for characterizing cardio-respiratory interactions generally, and their specific application in the infant population, might include the following:

https://doi.org/10.1109/TBME.2014.2323131

https://doi.org/10.3390/e23080939

 https://doi.org/10.3389/fphys.2020.01095

https://doi.org/10.1088/1361-6579/aac553

Author Response

Thank you for your suggestions. We added the following paragraph in the Discussion section:

"The current study represents a novel approach in understanding the interplay between bradycardia and CRC in preterm infants. Previous studies in this field, such as those by Faes et al. 2014 [46], Rozo et al. 2021 [47], Lucchini et al. 2020 [48], and Lucchini et al. 2018 [25], have laid important groundwork. Faes et al. study focused on the use of model-free tools for time series analysis to understand physiological system interactions, introducing a method to evaluate the direction, magnitude, and timing of information transfer between systems. Rozo et al. compared various methods to estimate Transfer Entropy (TE) in cardio-respiratory interactions, finding adaptive partitioning most effective. Their work emphasized the importance of choosing appropriate signals and methods for analyzing such interactions. Lucchini et al. 2020 study explored cardiorespiratory information transfer in healthy neonates, aiming to describe its development relative to gestational age. They extended the traditional TE measure to analyze both instantaneous and delayed effects between cardiac and respiratory systems. Lastly, Lucchini et al. 2018 investigated the phase coupling and its directionality in newborn infants, assessing the influence of gestational age at birth on the development of this synchronization.

The current study builds upon these foundations by specifically focusing on the differences in CRC during bradycardic and non-bradycardic events in very preterm infants. Unlike the previous studies which primarily involved healthy subjects or general neonate populations, this research delves into a more vulnerable group, very preterm infants experiencing bradycardia. The use of the PICS database for analyzing cardiac and respiratory time series, coupled with information theory measures, marks a distinctive approach. This study's findings about the lower entropy values in bradycardic segments and the implications for reduced complexity in cardiorespiratory dynamics during such events contribute significantly to the understanding of autonomic maturation and the interplay between cardiac and respiratory systems in this high-risk population. This direction is crucial for developing better diagnostic tools and enhancing healthcare outcomes for preterm infants."

New relevant references:

  1. Faes, L.; Marinazzo, D.; Montalto, A.; Nollo, G. Lag-Specific Transfer Entropy as a Tool to Assess Cardiovascular and Cardiorespiratory Information Transfer. IEEE Trans Biomed Eng 2014, 61, doi:10.1109/TBME.2014.2323131.
  2. Rozo, A.; Morales, J.; Moeyersons, J.; Joshi, R.; Caiani, E.G.; Borzée, P.; Buyse, B.; Testelmans, D.; Van Huffel, S.; Varon, C. Benchmarking Transfer Entropy Methods for the Study of Linear and Nonlinear Cardio-Respiratory Interactions. Entropy 2021, 23, doi:10.3390/e23080939.
  3. Lucchini, M.; Pini, N.; Burtchen, N.; Signorini, M.G.; Fifer, W.P. Transfer Entropy Modeling of Newborn Cardiorespiratory Regulation. Front Physiol 2020, 11, doi:10.3389/fphys.2020.01095.

Reviewer 2 Report

Comments and Suggestions for Authors

Authors answered all my questions. The manuscript seems sufficiently improved for publication

Author Response

Thank you. Your comments and suggestions have been instrumental in enhancing the quality of our work for publication."